# “An Aid with Soul”—Understanding the Determinants of Guide Dog-Owner Compatibility from Qualitative Interviews

**DOI:** 10.3390/ani13172751

**Published:** 2023-08-29

**Authors:** Yana Bender, Tim Matschkowski, Stefan R. Schweinberger, Juliane Bräuer

**Affiliations:** 1Max Planck Institute of Geoanthropology, DogStudies, Kahlaische Strasse 10, 07745 Jena, Germany; matschkowski@shh.mpg.de (T.M.); juliane.braeuer@uni-jena.de (J.B.); 2Department of General Psychology and Cognitive Neuroscience, Friedrich Schiller University, Leutragraben 1, 07743 Jena, Germany; stefan.schweinberger@uni-jena.de

**Keywords:** human–animal bond, guide dogs, dog–owner compatibility, personality

## Abstract

**Simple Summary:**

Guide dogs can help visually impaired persons to feel more confident and independent. Twenty-one guide dog owners reported the following factors to be important for a good match between a dog and an owner: sharing hobbies, similar activity levels or higher activeness in dogs, similar expressions of calmness; happiness; greediness; and friendliness. Owners also felt like a good match with their dog when they were both open or their dogs were more open than themselves and when they were dominant personalities and their dogs were more submissive. Moreover, the relationship to a former guide dog can have a big impact on the next relationship. Owners who felt similar in their personality to their dogs, as well as owners who felt like a good match with their dogs, reported positive aspects such as a strong bond and less influence from previous relationships. However, a strong bond might sometimes also have negative effects. The findings can help to understand what makes a dog and an owner a good match and improve the matching processes of guide dogs and handlers.

**Abstract:**

Guide dogs hold the potential to increase confidence and independence in visually impaired individuals. However, the success of the partnership between a guide dog and its handler depends on various factors, including the compatibility between the dog and the handler. Here, we conducted interviews with 21 guide dog owners to explore determinants of compatibility between the dog and the owner. Experienced compatibility between the dog and the owner was associated with positive relationship aspects such as feeling secure with the dog. Certain characteristics emerged as subjective determinants of compatibility, including shared hobbies, high levels of openness in both or only the dog, similar activity levels and higher activeness in dogs, similar expressions of calmness; happiness; greediness; friendliness; and a complementary dominance–submissiveness relationship. Owners who perceived themselves to be similar in their personality to their dogs often reported to have a strong bond, to feel secure with their dog and to be less influenced by previous relationships. However, our results suggest that a strong bond between the dog and the owner does not exclusively yield positive effects. Moreover, prior dog ownership seems to have a potentially strong impact on the subsequent relationship. Our results contribute to the understanding of dog–owner compatibility and may improve the matching process of guide dogs and their prospective handlers.

## 1. Introduction

The bond between humans and dogs is widely recognized as a unique and significant relationship [1]. Over the past decade, research has shed light on the remarkable cooperative tendencies displayed by dogs. Dogs show several prosocial behaviors, such as sharing food and informing, when they receive cues signaling the need for help [2]. This motivation for cooperation, paired with reward-based training, has fostered dogs helping humans in different areas of life, such as smelling scents of missing persons, assisting disabled people with opening doors and picking up objects or guiding blind people. However, those dog–human dyads are not always successful in their work, and neither are owners always satisfied with the relationship they share with their (working) dogs. This is reflected in the current high number of relinquishment of pet dogs in Western countries [3]. This is not only the case for family dogs but also for working dogs where the relationship between the dog and the handler may be unsuccessful. Lloyd and colleagues [4], for example, found that 36% of all dogs were returned to the guide dog training establishment before reaching retirement age (in a sample of *N* = 118 teams). This high return rate is associated with both economic and personal costs. The intensive training that the dogs have undergone is very expensive; assistance dogs can cost up to USD 50,000 (according to the National Service Animal Registry, 2019 [5]). In these dyads, where humans rely on and trust their dogs’ abilities to solve particular tasks (such as in assistance dogs but also police or rescue dogs), one can imagine that failure also may have tremendous social and personal consequences, including life-threatening ones in extreme cases. In addition to a sound education of the dogs and constant training, we here consider the matching of a working dog to its prospective handler as a factor for the development of functionality and satisfaction in a dyad [6].

### 1.1. Determinants of Success in Dog–Owner Dyads

The general literature background on the determinants of success in dog–owner dyads is still considerably small. Existing papers tend to stress uncontrollable aspects like the age and profession of the owner, the area of residence or the number of household members [7,8,9], which generally impact ownership in either a negative or a positive way. Aspects that can be used to actively form functioning dog–human dyads especially include personality aspects [10,11]. There is a substantial body of evidence suggesting that dog personality can be assessed along similar dimensions or traits that have been established to assess human personality traits [12,13]. In addition, we already know that the majority of dog–owner pairs resemble each other in their personality traits [14,15]. However, not much is known about the consequences this might have on functionality and satisfaction in the relationship. While it seems plausible that pairings of those similar individuals are especially functional, an alternative possibility could be that complementary traits favor the development of a functional team. Moreover, the particular attachment style of the dog and the owner, as well as the general quality of the bond, are factors that interact with these personality aspects and influence the performance of a dyad [10,16,17]. To date, no study has investigated these five potential aspects of a successful match (performance, dog and owner personality traits, quality of the bond and attachment aspects) and their interaction.

### 1.2. Functionality in Guide Dog–Owner Pairs

Besides this knowledge on general dog–human relationships, additional aspects might play a role when analyzing the success of working dogs and their handlers. The functionality of cooperation to solve different tasks determines their success on many levels, potentially also their bond and satisfaction within the general relationship. Thus, the importance of assigning the “right” dog to the “right” owner might even play a more significant role than in family dogs and their owners. Guide dogs are probably the type of working dogs that people trust their lives to the most, as reflected in their training, which lasts several months and, unlike for many other working dogs, usually is not performed by the owners [18]. While it is clear that the degree of compatibility between the human and the (guide) dog impacts the therapeutic value of the partnership [19], the few existing studies on the compatibility of guide dog–owner teams lack a viewpoint that includes detailed consideration of the personality traits of both counterparts [19,20,21]. Lloyd and colleagues [22] recently showed that within a sample of 50 New Zealand guide dog owners, the average subjective compatibility was very high. However, the determinants of this compatibility remain largely unexplored. Problems in research on guide dogs resemble those on family dog–owner matching (see above), including a small volume of research. Existing studies emphasize contextual factors and mobility factors [6,23]. Mobility is certainly important and is already considered by guide dog trainers; even evaluated processes to match guide dogs to their handlers, like the *Orientation and Mobility Outcomes (OMO)* tool, exist. [20]. Concerning contextual factors, a recent study has identified four essential areas: societal, social support, environmental and personal factors [23]. Many of these factors were outside the handler and the guide dog organization’s control, emphasizing the importance of research focusing on more modifiable issues of the dog–human relationship.

The widespread belief that unsuccessful relationships are caused by aspects of the dog (e.g., inadequate training or behavior in general) can be disproved by the observation that mismatched dogs can often be successfully rematched [4]. This suggests that a successful relationship depends on the interaction between the handler and the dog, instead of solely being one party’s fault. We therefore focus the interviews in the current study on the subjective fit that handlers feel between themselves and their dogs, thus exploring the potential power of matching processes. Our focus furthermore lies on aspects other than immutable contextual or already extensively studied mobility factors, such as personality traits, which represent a promising approach based on family dog research. Another aspect that will be considered is that former relationships have been shown to impact the subsequent relationships with the next guide dogs [24]. As an additional point, we included the potential influence of an existing or non-existing dog affinity (meaning that persons have general positive feelings toward and a preference for dogs) when acquiring the guide dog, which has not yet been investigated.

### 1.3. Research Gap and Study Rationale

A functional relationship between the owner and the guide dog increases subjective confidence and independence in blind persons and leads to better social relationships [25]. Consequently, there is a high relevance of research on what favors this functionality, especially in processes that are influenceable, such as the matching process. Compatibility between dogs and humans in these dyads is not yet well researched, as discussed above. There is a specific lack of studies that focus on the personality of both partners. Our aim is therefore to find determinants of compatibility between the dog and the owner. In order to do so, we conducted semi-structured interviews with guide dog owners. Guide dog owners often have many years of experience with different dogs and can be expected to have good experiential knowledge regarding the determinants that have affected the quality of different relationships. The methodological approach of conducting semi-structured interviews furthermore allows for freely exploring other psychological determinants of compatibility. 

## 2. Materials and Methods

### 2.1. Participant Recruitment and Demographics

We conducted expert interviews with 21 guide dog owners on what makes guide dogs for the blind and their owners a compatible, successful team. The interviews were realized in the setting of an extensive quantitative study whose subjects also include guide dog owners. We recruited participants through expert contacts (assistance and guide dog trainers, club chairmen) of the DogStudies Lab at the MPI GEA in Jena, who spread word of the study via newsletters, WhatsApp and Facebook groups, as well as calls in the local press. Participants had to be visually impaired or blind guide dog owners over 18 years old. After interested parties had contacted us by mail, a telephone appointment was arranged. The interviewer (Y.B.) explained the procedure and the possibility of participating in an interview. Of the 24 participants in the quantitative project, 21 owners agreed to take part and were interviewed in German between July 2022 and January 2023. Men constituted 19% (*N* = 4) of the sample, and ages ranged from 28 to 69 years (*M* = 54.1 years; for demographical details, see Table 1).

### 2.2. Ethics

The study received approval by the Max Planck Ethics Council on 27 June 2022 (Application No: 2022_12).

### 2.3. Realization of the Interviews

All interviews started with an introduction by the interviewer. She explained that the topic will be the relationship between dog and human and the subjectively experienced match between the two. Furthermore, she stated that the interviewee could give their information in as much detail or as concisely as they felt comfortable. A privacy statement was signed at the beginning of the quantitative behavioral study. We recorded the interviews via the Apple app “Voice Memos”. Twelve of the interviews could be realized in face-to-face live setting, while nine had to be conducted via telephone due to logistic and time reasons. The interview length ranged from 12.20 min to 45.15 min, with an average time of 25.19 min.

### 2.4. Questionnaire Design

Based on the state of research and our background knowledge, we focused on four aspects:The subjectively experienced match with their dog (based on personality traits);Relationship parameters such as satisfaction, initial expectations and problems;The comparison to and influence of other (former) dog–human relationships;The influence of dog affinity.

According to Kuckartz [26], we tried to balance the semi-structured questionnaire between questions based on the state of research and an open question format, to profit from the participants’ expertise and not influence them in their answers. This approach resulted in a minimum of nine and a maximum of twelve questions, depending on whether participants had owned another guide dog before and experienced problems within the relationship (see Table 2). The initial questions were more general and subsequently got more detailed, but note that the order could also be varied according to the interview flow, due to the semi-structured format. At the end of the interview, the participants also had the opportunity to add further aspects relating to the topic they felt were important. The interviewer sometimes asked more detailed questions about the differences between off- and at-work relationship, as major differences in these distinctive settings exist [27].

### 2.5. Data Processing

All interviews were transcribed via the software f4transcript [28]. Afterward, transcripts were manually checked and adapted according to Kuckartz [26]. For all analyses, the original German language was kept. The data were evaluated using qualitative content analysis according to Kuckartz [26], which ties in with Mayring’s content analysis [29]. The individual phases and the illustration in the form of a cycle enable a comprehensible and clear data evaluation (see Figure 1). Qualitative content analysis is characterized by the formation of categories that are worked out both inductively and deductively. Throughout the entire process as shown in Figure 1, an iterative and cyclical approach was employed to ensure a comprehensive and high-quality analysis of the interview texts. 

T.M. coded all interviews two times, firstly based on the a priori evaluated codes based on the questionnaires. T.M. and Y.B. then discussed the sub-codes based on the participants’ answers and defined the coding criteria. T.M. coded the data a second time, and afterward, Y.B. went through all of them again. Differing cases (about 20) were discussed within the research team. Table 3 shows the eight main and 13 sub-codes that resulted from this approach. All text passages relevant to the research question were assigned to one of these codes.

The analyses were again performed against the background of the methodology by Kuckartz [26] and with the analysis software f4analyse [31]. For this, the following analyses were carried out and written up in a logical order: category-based evaluation of main categories, correlations of subcategories within a main category and correlations between main categories. Besides analyses between the categories, other demographic aspects and shared features were considered in the next analysis step as suggested by Kuckartz [26]. 

## 3. Results and Discussion

### 3.1. Compatibility—Most Owners Feel like a Good Match for Various Reasons

Table 4 presents an overview of the main results of compatibility. Nineteen of twenty-one owners felt they were a good match with their dog, whereas two owners felt themselves and their dog were rather not a good match. Out of the 19 owners with a well-matched dog, most felt they and their dog were similar in their personality (14/19), and about one-quarter of them felt they were different (5/19). Of all participants, six owners (6/21) reported their dogs would adapt to them in some aspects. Four owners (4/21) reported their relationship with their dog was “love at first sight”, all of them continued to believe they were a good match (4/4), and most of them felt more similar in their personality (3/4). These four participants also reported that their expectations for their guide dog were met or exceeded.

Three owners (3/21) did not name any specific personality traits that they identified as matching or not matching with their dog. Research classifies these difficulties as a common phenomenon and that assessing one’s own personality is biased by egoistic and moralistic biases and is not always easy, or accurate [32,33,34]. Most of those owners who could not name precise personality traits (2/3) did not feel like a good match with their dog.

Many owners reported that they share certain characteristics that make them and their dog a good match. These read as follows: both are friendly (mentioned by three), greedy (mentioned by two), happy (mentioned by two), reserved, interested, cuddly, humorous, physical, loyal, scattered or both show no pain. For example, participant 6 said “My dog is friendly, greedy, and likes to swim. All characteristics that I have as well.” Comparable to the result regarding friendliness and happiness, Bauer and Woodward [16] found an expression of the trait “warmth” to predict owners’ attachment to the pet and satisfaction with the human–animal bond. Moreover, a study by Curb et al. [14] supports these results, in which four out of eight similarity characteristics between the owner and the dog were associated with owner satisfaction. Similarly, strong openness resp. extraversion in both was important for six participants. Four of them said they and their dog were open and approached other people together, while two described that they themselves approached other people openly, and their dogs did the same with other dogs. Participant 18 described it like this: “When someone comes up to us in town, he first approaches everyone in a friendly manner. He also offers his friendship to every dog. And that’s actually how I am, too.” Indeed, research supports the importance of openness in owners—it was found to be generally associated with greater attachment between the dog and the owner [35]. Two owners described that their dogs’ openness was a door opener to social contact with others for them, as they tended not to be open themselves, and another said it was a good fit that her dog was more open and curious than she was. This positive impact of high openness in dogs on relationship satisfaction has been observed by Cavanaugh and colleagues [36] before. Also, an explanatory approach can be found within the social support hypothesis [37], which proposes that companion animals act as facilitators of social interactions between other human beings and provide social support themselves [38]. If the dog expresses an open attitude toward other humans, this effect could be reinforced. 

In terms of complementary personality traits, the following results emerged. Two participants said they had the dominance their dog needed. This also is supported by previous research as Bauer and Woodward [16] reported the combination of submissiveness and dominance to be linked to higher attachment of the dog and the owner. Another participant reported that his dog and he complemented each other well because his dog grasped situations more quickly than he did. One owner described that she herself was often nervous and that her dog’s calm nature helped her to calm down more quickly. Similarly, participant 21 reported that her dog’s calm manner helped her overcome her anxiety: “Well, I am naturally a very fearful person (...) but (name of the dog) was supposed to show me that it can be done differently. And we achieved that.”

As a non-matching trait, one participant mentioned that her dog was too meticulous, while another reported that she was very fond of music and that her dog disliked music; both participants still felt like an overall good match with their dog. One owner who felt she was not a good match with her dog (1/2) indicated that her dog was too temperamental for her, while the other owner who felt she was not a good match could not provide specifics.

For an overview of temperament and activity level, see Table 5. Five participants felt they were a good match with their dog because they were similarly active and three because they were similarly calm. Three of all participants stated it was a good fit that their dog was more active than they were and thus carried them along (participant 10: “On days when I’m in a worse mood and maybe want to go for smaller walks only, he goes: ‘No, but we still have to do more. That’s good for you’, (…) Yes, he then brings you back on the right track.”). This has also been the case in a study by Chopik and Weaver [39], in which owners reported higher relationship quality when dogs were more active than themselves. These three participants also feel that themselves and their dog are generally different, with one of them not feeling like an overall good match with her dog. 

Participants reported the importance of shared hobbies within their relationship. Two owners said they and their dogs equally loved water. Another participant loved ball sports as much as her dog. Three owners felt the shared hobby of walking was a great fit, and two participants felt it matched their active lifestyles that their dogs could go everywhere with them and were always relaxed. One participant reported the matching common weather preference for winter. This is supported by former studies that identified shared hobbies to increase owner satisfaction [14,40] and decrease risk factors of problematic dog behavior [39].

### 3.2. Relationship Parameters—Can Dog and Owner Have a Too Close Bond?

Table 6 shows an overview of the results regarding relationship aspects. No owner reported being generally unsatisfied. Fourteen of all participants reported to have a subjectively strong bond with their dog (this was not asked explicitly but considered if owners stated how intimate/intense/close the relationship was or that the dog was their best friend/partner). From this group, most owners felt similar to their dog (12/14), and all felt they were a good match with their dog (14/14). Five participants felt very secure with their dog and stated that they could rely on them. All of these (5/5) had a strong bond and felt similar to their dog. Out of six teams in which the dog and the owner shared a high expression of the openness trait, almost all (5/6) had a strong bond. 

A variety of statements were made about the value of the dog, such as feeling dependent on it or seeing it as a family member. Participant 3 described the importance of her dog as follows: “(...) aid with soul. So, of course, a family member, friend, but just also an aid.” Three participants said their dogs brought a lot of relief and freedom in everyday life; two said they were partners and best friends. Two owners described that even in times when things were not going well with the dog, they still did not want to give it up. Two participants described a very close proximity to their dogs and that they had nothing else and found their lives no longer worth living without them (participant 20: “If he’s no longer there, then... then you can put me in the coffin right away. There is nothing more then.”). 

Previous studies have shown that benefits of pet ownership show when owners are moderately attached to their dogs [41,42]. Very high or extreme expressions of attachment are associated with the development of mental health issues (at least in elderly women, which are also strongly represented in our study [41]). Three of the strongly attached owners also felt negative social impacts on their life, and two owners found their lives no longer worth living without their dogs. These findings and the fact that we did not investigate the general satisfaction with life or mental status of our participants lead to the conclusion that negative psychological impacts on strongly attached owners cannot be excluded and need to be further explored. 

### 3.3. Strong Bonds Can Develop Even if Expectations Are Not Met 

Nine participants (9/21) indicated that the expectations they had for their future guide dog were met (for an overview, see Table 7). Three participants (3/21) said they had no expectations. Five (5/21) stated their expectations were even exceeded, and they all now share a strong bond with their dog. Two participants stated that their expectations had not been met. While existing research underlines the negative consequences of high expectations, such as increased returning rates [43], these two participants in our sample still have a strong bond with their dogs. One participant was ambivalent about whether her expectations were met. In addition, participants expressed expectations on the levels of mobility and, in private, fears about not being a good match and expectations of the general abilities of a guide dog in first-time owners. Participant 11 said, “As someone who has never had a guide dog (...) you don’t have expectations, but you have dreams about what the dog could do. But it was really hard work (...) where I thought to myself in between: “Yes, why do I need the guide dog, if I have to do everything myself anyway?”.

### 3.4. Former Guide Dog Relationships Can Have Strong Influence on the Subsequent Ones

Of 12 participants who had one or more guide dogs before their current one, three stated they were just as well matched with the previous one as they were with their current one (for an overview of compatibility in previous relationships, see Table 8). None of these (0/3) were disappointed in their expectations of a guide dog, and two (2/3) had a strong bond with their dog.

One participant reported that her previous dog was a better match for her because he was more generally sensitive and temperamental when playing, and they were generally more similar. Participant 4 reported that she had a closer bond with her previous dog because “I sometimes felt myself that he notices and thinks what I feel and when I wasn’t feeling well or I was upset or something, he felt and sensed that exactly.” These two participants who experienced their previous dog as a better fit are the ones who experienced their current dog as a mismatch, did not have a strong bond or felt very secure with their dog. Participant 2 described it as follows: “So you always have a sweetheart dog, unfortunately, and I’m sorry about that, that was just her predecessor. That’s maybe like first love too.” This might also explain why they could not name precise personality traits as determinants of compatibility: The individuals may have had certain expectations about the compatibility based on their previous relationship. However, when faced with the actual interactions and behaviors of their dogs, they may have found discrepancies in comparison to their former dog. This incongruence could lead to difficulties in identifying and reporting specific traits as determinants of compatibility. It is possible that the high expectations due to former relationships themselves influenced the feeling of incompatibility. A study by Powell and colleagues [44] furthermore showed that previous dog owners had reduced odds of expecting challenges than non-owners. This indicates bias through selective recall of positive experiences from previous ownership. The same could be the case for the two interviewees in our study. What should be mentioned when looking at this possible explanation, though, is that none of the two mismatching owners reported to be disappointed in their expectations (one was ambivalent though, and both described their answer to the question more on a mobility level).

Two participants (2/12) reported their previous dog was very active, which suited them well at the time; they are now somewhat calmer, and the current, less temperamental dog is a better match. Five owners (5/12) reported their current dog was a better match than previous ones, one because his calmness suited her, one because the chemistry was just better, one said the previous dog was too reserved, and another said he had too much anxiety. One participant felt her previous dog was unsuitable for her but believed this was also due to his training with punishment, which she herself did not use. Participant 13 reported that his previous dog was good for beginners and his recent one a little too temperamental: “So I no longer need to be completely carefree. But I don’t necessarily need quite as much stress as with this one right away. Something in between would be quite good”.

Four prior guide dog owners (4/12) felt their current relationship was influenced by their previous one (for an overview, see Table 9). Two of them (2/4) felt the match with their current dog was better, and two (2/4) felt it was worse than with their previous one. The fact that none of those owners thought the dogs equally matched might hint at a lack of neutral evaluation due to an extremely positive or negative prior relationship. This underlines the power of the influence of prior relationships as already found by Lloyd and colleagues [24].

Five owners (5/12) said they were not affected by the relationship with their previous guide dog, of those only one (1/5) felt any impact on their social life, and none (0/5) were disappointed in their expectations of a guide dog. Two (2/12) participants did not give specific statements about being influenced, and one person was ambivalent. There was no apparent connection between being influenced and second-, third- or fourth-time ownership. Two individuals said their expectations had been very high due to good previous relationships; these two were also the two participants who experienced themselves as not matching with their dog. For these two participants, also a higher attachment to their prior dog could be the case that led to more grief and sorrow, in turn negatively influencing the subsequent relationship [45]. One participant said she had known exactly what she did not want because of a previous mismatch. Four participants (4/12) said it was important to them not to compare the dogs to each other. Two owners (2/12) said the transition was difficult for them because the dogs had very different personalities. One first-time owner commented that she would have liked to have had some dog experience and believed this would have positively influenced the current relationship.

### 3.5. Dog Affinity Is Not Crucial for Compatibility

To our knowledge, no study has yet analyzed the impact of dog affinity on success in (guide) dog–owner relationships. It has been shown that perceived cuteness (which might be higher in dog-affine persons) predicts the relationship quality though [46]. In our sample, sixteen participants (16/21) said they had an affinity for dogs before getting their guide dog (for an overview, see Table 10). Five owners (5/21) said they had no previous connection to dogs, were afraid of dogs or would not have acquired a dog but for their disability. Eleven owners (11/21) had a family dog before their first guide dog. One participant grew up with dogs but would not have acquired a dog without her visual impairment; this was counted in the second group of non-dog-affine participants. Two owners expressed the opinion that one should only get a guide dog if one would otherwise acquire a dog. Of the five owners who said they had not been dog-affinitive before, all felt like a good match with their current dog, and three of them (3/5) had a strong bond with their dog. Being non-dog-affine therefore did not seem to negatively impact the relationship. Moreover, even a potentially positive aspect emerged: even though four owners (4/5) had another guide dog before, none of them (0/4) felt influenced by their previous relationship.

### 3.6. Owners Experience Positive and Negative Effects on Their Social Life

Twelve participants (12/21) mentioned the effects of living with a guide dog on their social life. Of these, two owners (2/12) reported exclusively negative experiences, and one (1/12) reported both positive and negative experiences, such as the dog getting too much attention from others and rejection of the dog (participant 7: “And there is also sometimes a lot of rejection in my private environment, where I (…) thought, was it right (to get the dog)?”). The most frequently mentioned positive effect was increased social contacts. For example, participant 16 reported, “When you have a dog and you’re a little more open-minded, you have a lot of friends who have dogs.” This is supported by literature that suggests that dog ownership can increase opportunities for social contact and even new friendships [47], as well as specifically increase and change social interaction for guide dog owners [25]. Of the twelve owners who experienced impacts, nine had a strong bond with their dogs, including the three persons who experienced negative impacts of the social environment. This reflects that negative consequences do not necessarily impair the relationship. The negative social consequences might even be a result of the potentially “too close” bond itself, as discussed above.

### 3.7. Problems in Everyday Life Are Diverse

Nineteen of twenty-one participants reported problems in everyday life with their dog. Most of these problems were undesirable characteristics of the dog, such as greediness (named by ten participants) and being too open to other people (named by four participants). The other problems mentioned were diverse, including allergies of the dogs, yapping and over-excitement. The two participants who did not name any problems felt that they and their dog were a good match, and one shared a strong bond with her dog.

### 3.8. No Meaningful Differences in Off-/At-Work Relationship

Although inquired at some points by the interviewer, the owners reported only little difference between the at-work and off-work relationship throughout their interviews. Based on theory, it was expected to find major differences between those two modes and that the working relationship has a bigger impact on whether or not handlers consider the match to be a successful one [22]. We even found contrary results: both owners who felt like a mismatch with their dogs reported that the guide work was well functioning.

## 4. What Is Better—Similar or Different Teams?

Within the comparison of different and similar teams, the prominent differences in the following areas evolved: sharing a strong bond, feeling secure with the dog, being influenced by the previous relationship and feeling the current dog matches better than the previous one. As shown in Table 11, the difference in compatibility between different and similar teams was not striking. One participant in each group felt like a mismatch with their dog.

Our results show that similarity in ownership is not clearly indicative of higher compatibility but is associated with positive relationship characteristics (the similar teams more often shared a strong bond, felt secure with their dog, felt their dog matches better than the one before and were less often influenced by their previous relationship). This is also supported by the state of research. Studies showed several advantages of similar expressions of traits, such as more positive attitudes toward their dog when they were similar on the dimension of warmth [48] and owner satisfaction when they were similar to their dogs in sharing possessions or the enjoyment of running outside [14]. But there are also studies that emphasize the positive impact of different combinations of traits, such as higher relationship satisfaction when dogs are more open, agreeable and neurotic than their owners [36]. The positive impact of being similar in the dimension of openness, as well as high openness in the dog, has also been described by participants in our study, as discussed above. Interestingly, a too high openness of the dog was one of the most often named problems of the participants, which probably plays a more dominant role in guide dogs than in family dogs, where they should not be distracted from external stimuli when guiding their owners. This discrepancy in our results, as well as in general research, underlines the need for extensive quantitative studies on the dog–owner compatibility based on their personality traits.

## 5. Limitations

Choosing a qualitative approach allowed us to freely explore possible determinants of guide-dog–owner compatibility and include owners’ subjective perspective. It needs to be mentioned, though, that this type of research always has interpretative parts [49] and contains the possibility of social desirability bias (a tendency to present reality as what is perceived to be socially acceptable [50,51]). Additionally, nearly half of the interviews were conducted via telephone. We did not see any systematic differences between the groups of persons interviewed via telephone vs. those interviewed in person in our study. Still, telephone interviews are discussed to have advantages such as cost and time efficacy but also disadvantages such as potential effects on the content of the responses due to anonymity created by spatial separation [52,53].

Another bias that probably occurred in our study is the volunteer bias [54]. Accordingly, unsatisfied owners who experience major problems in their guide dog relationship will be less likely to expose those in an extensive study. This could contribute to the fact that our sample only included a very small number of incompatible teams. Another characteristic of the sample that limits generalizability is the fact that mainly older and female owners, as typical for animal research [55], participated in the study.

## 6. Conclusions and Future Research

Owners who perceived themselves and their dogs as a good match were more likely to identify personality traits as determinants of compatibility, experience a strong bond and feel secure in their relationship with their dog. They also expressed satisfaction in their expectations and were rarely influenced by previous relationships, underlining the importance and positive potential outcomes of a good match.

Certain traits emerged as subjective determinants of compatibility, including shared hobbies, shared high openness and high openness in the dog, similar activity levels and higher activity in dogs and the combination of dominant owners with submissive dogs. Additionally, similarities in the expressions of calmness, happiness, greediness and friendliness were deemed important. While similar teams tended to have a stronger bond, feel more secure and be less influenced by previous relationships, the differences in compatibility between different and similar teams were not particularly salient. Our study therefore points toward a positive influence of similarity on the relationship, but future research is necessary to confirm this assumption.

Dog affinity, expectations not being met and differences between the on- and off-work relationship did not influence compatibility or other relationship parameters in our sample. Prior dog ownership, however, seems to have a potentially strong impact, as participants who reported to be influenced due to a former very positive guide dog relationship did not feel compatible with their current dog. The majority of owners reported positive social consequences associated with guide dog ownership, with only a few reporting negative effects. Nevertheless, all of them maintained a strong bond with their dog, indicating that these consequences did not impair the overall relationship. It is important to consider, however, that our results, in line with the existing literature, suggest that a strong bond between the dog and the owner does not exclusively yield positive effects.

Having taken advantage of a qualitative approach to freely explore possible aspects of compatibility, it is necessary to further study the parameters found. What is needed, then, is a comprehensive study that captures personality traits of participants and their dogs using validated questionnaires, thus circumventing the problem that many participants had difficulty assessing them freely. In addition, it would be important to further explore the differentiation between the off- and at-work relationship, as well as to include a measurable outcome in the area of mobility of guide dogs, such as an obstacle course. With respect to our results, it could also be revealing to further investigate the impact of high attachment between the dog and the owner on the owners’ mental status or consequences in their social life.

However, the above-named conclusions contribute to the understanding of dog–owner compatibility and can already be used to improve the matching process of guide dogs and their prospective owners. This furthermore can potentially increase the success rate of compatible matches and thus the animal welfare, as well as the mobility of visually impaired persons.

## Figures and Tables

**Figure 1 animals-13-02751-f001:**
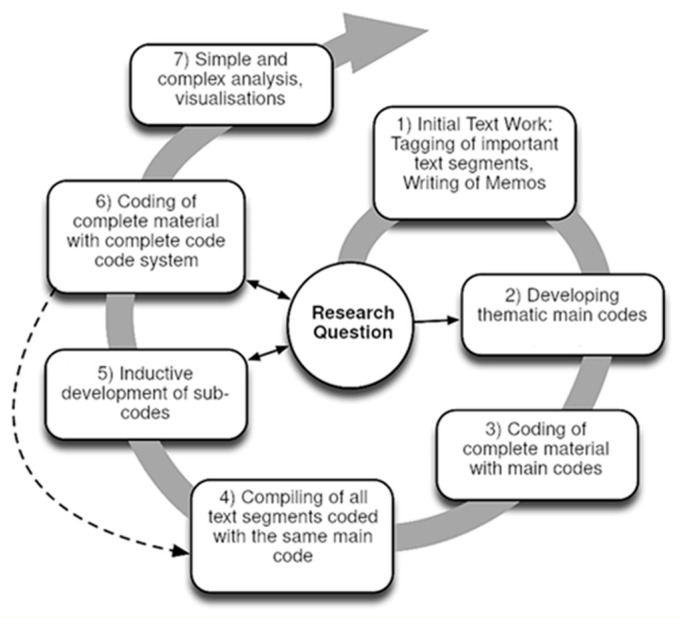
Process of a content structuring content analysis as presented by Kuckartz [30].

**Table 1 animals-13-02751-t001:** Participants demographics.

Number	Gender	Age (Years)	Impairment	Interview Location	Dog’s Age (Years)	Dog’s Sex	Time Together	Guide Dog History
1	F	57	Fully blind	Telephone	9	F	7 years	3rd
2	F	67	Fully blind	Telephone	8	F	7.5 years	2nd
3	F	47	Fully blind	In person	3	M	1.5 years	2nd
4	F	45	Partially blind	Telephone	3	M	1.5 years	3rd
5	F	69	Partiallly blind	Telephone	2	M	8 months	4th
6	F	36	Fully blind	In person	6.5	F	4.5 years	2nd
7	F	28	Fully blind	In person	4	F	2 years	1st
8	F	61	Partiallly blind	In person	7	M	5 years	1st
9	F	67	Fully blind	Telephone	10	M	8 years	3rd
10	F	51	Partially blind	In person	3	M	1.5 years	3rd
11	M	55	Partiallly blind	Telephone	4	F	2 years	1st
12	F	64	Partiallly blind	In person	5	F	3 years	1st
13	M	46	Partiallly blind	In person	8	F	6 years	2nd
14	F	36	Fully blind	In person	6.5	F	3.5 years	2nd
15	M	65	Partiallly blind	Telephone	5	F	2.5 years	1st
16	F	59	Partiallly blind	Telephone	2.5	F	11 months	3rd
17	F	51	Partiallly blind	In person	7.5	F	3.5 years	4th
18	M	59	Partially blind	In person	11	M	9 years	1st
19	F	56	Fully blind	In person	3	M	5 months	1st
20	F	65	Partially blind	Telephone	6	M	4 years	1st
21	F	52	Partially blind	In person	7	F	2 years	1st

**Table 2 animals-13-02751-t002:** Questions of the semi-structured interview.

**Question 1**	Please describe your relationship with your current guide dog and how satisfied you are with it.
**Question 2**	Do you think you and your guide dog are a good match?
**Question 3**	Which of your personality traits do you think are a particularly good match? Example: openness to new experiences, agreeableness, and extraversion (with short explanations).
**Question 4**	Which of your personality traits do you think are less compatible?
**Question 5**	Are there any characteristics you would change in your dog if you could?
**Question 6**	What problems do you experience in your daily life with your dog that have to do with your dog’s behavior or the relationship between you?
**Question 7**	Do you have any idea where these problems might be coming from?
**Question 8**	Have the expectations you had for your guide dog been met?
**Question 9**	Did you have another guide dog before your current guide dog, and if so, please tell me about your relationship with him.
**Question 10**	Do you feel you matched better or worse with your previous guide dog?—What could have been the reason for this?
**Question 11**	Do you feel that the previous relationship affected your current relationship?
**Question 12**	Before living with your guide dog, would you have described yourself as a dog person, or did you like dogs?

**Table 3 animals-13-02751-t003:** Main- and sub-codes used for coding the interviews.

**1 Compatibility**	**4 Compatibility with previous guide dogs**
1.1 Specific personality traits	4.1 Specific personality traits
1.2 General temperament	4.2 General temperament
1.3 General activity/energy level	4.3 General activity/energy level
1.4 General similarity	4.4 General similarity in personality
1.5 General difference	4.5 General difference in personality
**2 Bond and relationship aspects**	**5 Influence of previous guide dog relationship**
2.1 Positive attributes of the dog	**6 Dog-affinity**
2.2 Shared experiences with the dog	**7 Effects on social life**
2.3 Overall importance of the dog	**8 Problems in everyday life**
**3 Expectations and fulfillments**	

**Table 4 animals-13-02751-t004:** Overview of compatibility.

Good match: *N* = 19	Different: 5/19
	Similar: 14/19
Mismatch: *N* = 2	Different: ½
	Similar: ½
Love at first sight: *N* = 4	Good match: 4/4
	Similar: ¾
	Expectations met or exceeded: 4/4

**Table 5 animals-13-02751-t005:** Overview of temperament and activity.

Good match because both active: *N* = 5	
Good match because both happy: *N* = 2	
Good match because both calm: *N* = 2	
Good match that dog more active: *N* = 3	Different: 3/3 Overall Good Match: 2/3
Enjoying shared hobbies: *N* = 6	

**Table 6 animals-13-02751-t006:** Overview of quality of the bond.

Strong bond: *N* = 14	Similar personality: 12/14
	Good match: 14/14
Feeling secure: *N* = 5	Strong bond: 5/5
	Similar personality: 5/5

**Table 7 animals-13-02751-t007:** Overview of expectations.

Expectations met: *N* = 9	
No expectations: *N* = 3	
Expectations exceeded: *N* = 5	Strong bond: 5/5
Expectations not met: *N* = 2	Strong bond: 2/2

**Table 8 animals-13-02751-t008:** Overview of compatibility in previous relationships.

Good match before and now: *N* = 3	Expectations disappointed: 0/3
	Strong bond: 2/3
Better match before: *N* = 2	Good match: 0/2
	Strong bond: 0/2
	Feeling secure: 0/2
Better match now: *N* = 5	

**Table 9 animals-13-02751-t009:** Overview of influence of previous relationship.

Influenced by previous relationship: *N* = 4	Match with current dog better: 2/4
	Match with previous dog better: 2/4
Not influenced by previous relationship: *N* = 5	Influence on social life 1/5
	Expectations disappointed 0/5

**Table 10 animals-13-02751-t010:** Overview of dog affinity.

Dog affinity: *N* = 16	
No dog affinity: *N* = 5	Good match: 5/5
	Strong bond: 3/5
	Influenced by previous relationship: 0/4
Owned/lived with family dog before: *N* = 11	

**Table 11 animals-13-02751-t011:** Prominent differences in the comparison of different and similar teams.

Aspect	Percentage in Different Teams (*N* = 6)	Percentage in Similar Teams (*N* = 15)
Good match	5/6 (83.3%)	14/15 (93.3%)
Strong bond	2/6 (33.3%)	13/15 (86.7%)
Feeling secure	0/6 (0%)	5/15 (33.3%)
Influenced by previous relationship ^1^	3/5 (60%)	1/7 (14.3%)
Match with current dog better ^1^	1/5 (20%)	4/7 (57.1%)

^1^ in participants that owned guide dogs before (*N* = 5 in different teams, *N* = 7 in similar teams).

## Data Availability

The data presented in this study are not publicly or otherwise available due to the need to protect the anonymity of the participants.

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
