# Peer review of "“An Aid with Soul”—Understanding the Determinants of Guide Dog-Owner Compatibility from Qualitative Interviews"

_animals, 2023, doi:10.3390/ani13172751_

Round 1
Reviewer 1 Report
This paper describes results of a qualitative study aiming to understand whether certain personality traits, or other factors, are associated with success of guide dog/handler pairings. It has the potential to be an important contribution to the field, but it needs some adjustments to improve clarity and flow before I can recommend it for publication.
Major criticisms
First, the results meander considerably, and it appears that the authors are attempting to take qualitative data and turn it into quantitative data. This is neither instructive nor appropriate, and it means that the key messages of the findings are getting lost. It is not clear what actually matters, and what is less important. By presenting the findings are they are currently, there is information overload and no signpointing about what we need to be focusing on. To remedy this, I have a couple recommendations:
1. combine the Results and Discussion. Currently, the findings are reported and then basically rehashed in the Discussion. Qual research is especially amenable to combining these two sections to improve flow. That way, it is possible to report an important finding, and talk about what it means in the context of the larger body of literature, all at the same time, reducing repetition.
2. decide the most important outcomes and focus on those, rather than presenting ALL of the findings. It's not necessarily wrong to have 10 key themes, although that is a large number, and reporting/discussing all of them means that the most important elements of the study are being lost. I recommend instead omitting the less relevant aspects and just picking 3-5 key outcomes instead. The 'most important' may be the ones that the largest number of participants mention, or perhaps they are the most theoretically robust as determined by the authors. It doesn't really matter which constraints apply, as long as these are described in the methods section. If the authors are especially interested in personality, then focus on that. Since that didn't really get an instructive response from the participants, though, maybe pick something else that tells a useful story: perhaps the aspects of the relationship that can be controlled, as the authors mention this in the text. This will enable the findings and discussion to be smaller and tighter, and the key messages will shine through.
Second, the methods section needs some more information. What sort of reliability checks/bias reduction were applied for the analysis? Was there cross-coding? Or extensive discussions among the research team in an iterative fashion? The authors mention the Kuckartz guidelines, which is useful, but a bit more detail is required on this point.
Finally, the Table 3 with the codes and subcodes should be in the results section, not the analysis. Presumably these codes are an outcome of the analysis, although I appreciate the circular nature of qual analysis. Nonetheless, they were not applied a priori, as I understand it, so they should be presented as results. Also, I suggest altering the table somewhat so that there's an illustrative quote for each code/sub-code. That will enhance readability.
Other comments
Simple summary and abstract - instead of just mentioning that certain factors had an impact, be sure to note the direction of impact. If the direction is dependent on other variables, mention that, too. Since most people will unfortunately only read the summary and abstract, the key messages should always be included in these. If the authors streamline the findings, though, these will probably need to be updated anyway.
Keywords - the word 'interviews' doesn't need to be in the keywords because it's already in the title. See this website for recommendations on writing keywords, an increasingly important part of the publishing process given the field's current emphasis on systematic reviews: https://getproofed.com.au/writing-tips/how-to-pick-the-best-keywords-for-a-journal-article/
Introduction
L45 - this is related to pet dogs, not working dogs, even though the rest of the para is about pet dogs. Suggest removing or amending language for clarity
L51 and 53 - suggest changing 'service dogs' to 'assistance dogs' per the recommendation in a recent paper: Howell, Tiffani J., et al. "Defining terms used for animals working in support roles for people with support needs." Animals 12.15 (2022): 1975.
L85 - change 'usually cannot be done by the owners' to 'usually is not done by the owners'. It can be done in some cases and it depends on the jurisdiction and, of course, the owner's own abilities. There is no specific aim statement in the Intro. There are several sentences that describe the intention for the paper, but they are all slightly different. Suggest adding a discrete aim at the end of the intro. Methods L133 - suggest changing 'spread the study via..' to 'spread word of the study via...' Table 1 - what does 'rest of sight' mean? Table 1 - in presence should be in person Table 1 - add (years) to Dog's Age column L147 - suggest changing 'introduction on the side of the interviewer' to 'introduction by the interviewer'. L157 - suggest changing 'own thoughts' to 'our background knowledge' L161 - could the authors please clarify 'dog affinity'? It's not clear exactly what it means in this context, and particularly how it differs from bond with the guide dog. Table 2 Question 3 - were these the terms used? Did they require any explanation? Perhaps these terms were the reason why the authors didn't get much good info about personality? They are quite technical and not usually the sorts of words people would use unless they have a psych background, at least among English-speakers. People tend to know what introverts and extraverts are, but things like openness to experience and agreeableness are less immediately clear. L176 - should 'controlled' be 'checked'? Table 3 - could the authors please briefly expand on what is meant by 'general similarity' or 'difference'? Results Section - suggest changing header to Findings L220-226 - would this be more related to openness or extraversion? It sounds more like the latter to me. L240 - why was this participant not prompted for more detail on this question? It seems very important. Table 5 - what does 'matching that dog more active' mean? Please clarify. It is made clear in text but tables and figures should stand alone and not require further in-text clarifications Table 6 - in 'similar' - does this mean similar in personality or something else? L290 - 'previous ones' are these previous Guide dogs? previous dogs? previous pets? Clarify L290 - Four out of how many participants? Table 7 - again, please clarify the difference between dog affinity and bond. L319 - what does 'a lot of temperament' mean? L342 - 'these relationships' - current dog or previous dog? Table 10 - should 2/2 in the first row be 2/4? L373 and 375 - 'dog-affine' and 'dog-affinitive' are not commonly used in English. Suggest changing to 'having an affinity with dogs' L381-383 - what about the other 9? They aren't mentioned. L382-383 - suggest moving the clause 'and one reported both positive and negative experiences' below the examples of negative experiences, for clarity. L384-385 - this quote is unclear. I don't know if it's the translation or something else, but I can't understand what it means. Could the authors please clarify? Tables 12 and 13 - suggest adding the Ns to each row as well as the % if the authors decide to keep these tables. I would suggest removing most of the tables in the results section that try to assign quant values to the findings, though. L527-529 - does this lead to, or is it associated with? I believe this study was cross-sectional so causal inferences should be avoided. Conclusions L632 - add an example of relevant personality traits L657 - I wouldn't say that the qual approach has been 'exhausted'. There's still plenty of qual work that could be done in this space. Suggest removing. L663 - 'parkours', I believe, is the wrong word here. In English it's usually conceptualised not just as getting from A to B most efficiently, but it's become a sport for jumping around (e.g., up and down stairwells). Therefore, I would suggest changing to '...such as efficiency of guide dogs vs other mobility aids' or similar.
English is fine. I made a few suggestions in the comments above where it could be improved slightly, but mostly it's good.
Reviewer 2 Report
A very well-presented manuscript on an important topic.
I have only a few minor queries / comments.
1. In introduction when you say "Dogs show several prosocial behaviors, such as sharing and informing..." I am unclear what sharing means in this context.
2. lines 105-106: "We therefore focus the interviews in the current study on the subjective fit that handlers feel between them and their dogs" should be "We therefore focus the interviews in the current study on the subjective fit that handlers feel between themselves and their dogs"
3. line 161: the term " dog-affinity " (the last bullet point) should be defined for clarity (is it simply liking dogs or something else?)
4. line 337: where you say "and his recent one a little to temperamental" do you mean to say "and his recent one a little too temperamental:"?
5. in line 459, "One possible explanation for this might lie within the fact, that..." the comma is unnecessary
6. in line 467, "It is also thinkable, that.." should be phrased as "It is also possible that..."
7. lines 550-551, where you say: "Though it is known, that perceived cuteness (which might be higher in dog-affine persons) predicts the relationship quality" this might be better expressed (and not a sentence fragment) as : "It has been shown that perceived cuteness (which might be higher in dog-affine persons) predicts the relationship quality"
8. line 574 - should this be "“too close” bond itself" instead of "“to close” bond itself" ?
9. line578 - where you say "These two maybe contradictive seeming results..." this might be better expressed as "These two apparently contradictory results..."
10. It was reported as a limitation, but were there any SYSTEMATIC DIFFERENCES in the dat between those interviewed in person and those interviewed over the phone? Even in terms of interview length?
I notes only a few minor instances where rewording would be useful.
Round 2
Reviewer 1 Report
This manuscript has been tightened up and improved considerably. I am happy to accept it for publication.